# DNA Electrochemical Biosensor Based on Iron Oxide/Nanocellulose Crystalline Composite Modified Screen-Printed Carbon Electrode for Detection of *Mycobacterium tuberculosis*

**DOI:** 10.3390/molecules25153373

**Published:** 2020-07-24

**Authors:** Mohd Hazani Mat Zaid, Che Engku Noramalina Che-Engku-Chik, Nor Azah Yusof, Jaafar Abdullah, Siti Sarah Othman, Rahizan Issa, Mohd Fairulnizal Md Noh, Helmi Wasoh

**Affiliations:** 1Department of Chemistry, Faculty of Science, Universiti Putra Malaysia, Serdang 43400, Malaysia; zani@ukm.edu.my (M.H.M.Z.); azahy@upm.edu.my (N.A.Y.); jafar@upm.edu.my (J.A.); 2Department of chemical sciences, Faculty of Science and Technology, Universiti Kebangsaan Malaysia, Bangi 43600, Malaysia; 3Department of Bioprocess Technology, Faculty of Biotechnology and Biomolecular Sciences, Universiti Putra Malaysia, Serdang 43400, Malaysia; engkuamalina@gmail.com; 4Institute of Advanced Technology (ITMA), Universiti Putra Malaysia, Serdang 43400, Malaysia; 5Department of Cell and Molecular Biology, Faculty of Biotechnology and Biomolecular Sciences, Universiti Putra Malaysia, Serdang 43400, Malaysia; sarahothman@upm.edu.my; 6Bacteriology Unit, Infectious Disease Research Centre, Institute for Medical Research, Jalan Pahang, Kuala Lumpur 50588, Malaysia; rahizzan@imr.gov.my; 7Cardiovascular Diabetes and Nutrition Research Centre, Institute for Medical Research, Jalan Pahang, Kuala Lumpur 50588, Malaysia; fairul@imr.gov.my; 8Halal Product Research Institute (IPPH), Universiti Putra Malaysia, Serdang 43400, Malaysia

**Keywords:** *Mycobacterium tuberculosis*, iron oxide, nanocellulose crystalline, cetyl trimethylammonium bromide, DNA biosensor

## Abstract

Death from tuberculosis has resulted in an increased need for early detection to prevent a tuberculosis (TB) epidemic, especially in closed and crowded populations. Herein, a sensitive electrochemical DNA biosensor based on functionalized iron oxide with mercaptopropionic acid (MPA-Fe_3_O_4_) nanoparticle and nanocellulose crystalline functionalized cetyl trimethyl ammonium bromide (NCC/CTAB) has been fabricated for the detection of *Mycobacterium tuberculosis* (MTB). In this study, a simple drop cast method was applied to deposit solution of MPA-Fe_3_O_4_/NCC/CTAB onto the surface of the screen-printed carbon electrode (SPCE). Then, a specific sequence of MTB DNA probe was immobilized onto a modified SPCE surface by using the 1-ethyl-3-(3-dimethylaminopropyl) carbodiimide/N-hydroxysuccinimide (EDC/NHS) coupling mechanism. For better signal amplification and electrochemical response, ruthenium bipyridyl Ru(bpy)_3_^2+^ was assigned as labels of hybridization followed by the characteristic test using differential pulse voltammetry (DPV). The results of this biosensor enable the detection of target DNA until a concentration as low as 7.96 × 10^−13^ M with a wide detection range from 1.0 × 10^−6^ to 1.0 × 10^−12^ M. In addition, the developed biosensor has shown a differentiation between positive and negative MTB samples in real sampel analysis.

## 1. Introduction

*Mycobacterium tuberculosis* (MTB) is an infectious bacterium responsible for tuberculosis (TB) pulmonary disease [1]. In 2018, around 10.0 million people were estimated to have positive symptoms related to TB (WHO, 2019). According to the report, TB affects people of both sexes of all ages. The largest group involves men rather than women (aged ≥ 15 years), who accounted for 57% of all the cases. Among all TB cases, 8.6% were people living with HIV [2,3]. Various efforts have been carried out to improve TB diagnosis to reduce the number of deaths as set by the World Health Organization (80% reduction in TB incidence rate per year). These only can be achieved if the high-risk people are diagnosed correctly. Hence, the most effective treatment regimen should be started as early as possible. Over the past year, a complete medical evaluation for TB was based on laboratory diagnosis, majorly depending on the tuberculin skin test [4], chest radiography [5], acid fast bacilli (AFB) staining [6] and microbial culture [7]. These methods are accessible in developing countries; however, toxic reagents and more prolonged time of analysis are needed along with highly trained personnel for successful tests [8]. Furthermore, the implementation of nucleic acid amplification assays such as polymerase chain reaction (PCR) is limited by high cost and poor performance under extreme field conditions [9]. Therefore, there is a need to develop another fast screening method for in situ and real-time analysis.

Due to its considerable benefits, electrochemical means have become some of the most cost-effective analytical methods, particularly in sensing applications [10,11]. These advantages include rapid response, high sensitivity, low cost, and ease of use [12]. In recent years, electrochemical detection based on DNA hybridization exhibited excellent tools for DNA bioassay. Principally, this approach mainly relies on the response of current or resistance generated by the oxidation of guanine base and DNA intercalators [13]. Based on voltammetric methods, a metal complex has always been used as DNA intercalator for signal measurement between the electrode surface and DNA probes such as in the measurement of differential pulse voltammetry (DPV) and Squarewave voltammetry (SWV) [13,14]. Meanwhile, the use of a metal complex, especially tris(bipyridine) ruthenium (II), can be advantageous for the DNA electrochemical method to amplify the signal to obtain high sensitivity detection [15,16]. In order to increase the sensitivity of electrochemical biosensors, further modification is necessary to alter the surface of the electrode, particularly the conductivity of the electrode [17,18]. This can be achieved by tailoring the electrode surface with nanomaterials of high surface area and high catalytic activity with additional unique biocompatibility behaviour [19,20]. In this context, Fe_3_O_4_ offers tremendous potential advantages in the clinical and medical sensor, as reported in several studies [21,22]. Since Fe_3_O_4_ is highly hydrophobic, surface modification can be made through covalent bonding or physical adsorption using various biocompatible polymers [23]. Moreover, the high compatibility of Fe_3_O_4_ with other polymers offers a significant potential extent to improve the electrochemical performances of the respected biosensors. Nanocellulose crystalline (NCC) derived from biopolymer cellulose is a class of functional nanomaterial which exhibits promising potential for sensing application improvement. The NCC offers a plethora of outstanding properties, such as low toxicity, eco-friendly, high surface area, and good biocompatibility that can be functionalized easily by using different functional groups [24,25]. Previously, the NCC has been composited with polypyrrole [26], graphene oxide [27], gold nanoparticle [28], and has successfully become a scaffolding material for immobilization enzymes, DNA and other biomolecules to demonstrate the excellent synergistic effect that boosted electrochemical performance.

In this present study, we reported the development of electrochemical DNA biosensor based on Fe_3_O_4_ functionalized by 3-mercaptopropionic acid (MPA) through carboxyl (–COOH) functional group (MPA-Fe_3_O_4_) and followed by composited with nanocellulose crystalline/cetyl trimethyl ammonium bromide (NCC/CTAB) and, finally became MPA-Fe_3_O_4_/NCC/CTAB on screen printed carbon electrode (SPCE). To achieve this goal, negatively charged MPA-Fe_3_O_4_ and positively charged NCC/CTAB were incorporated through ionic interaction as a matrix for conductivity enhancement, promoting an electron transfer and subsequently act as an immobilization platform for DNA. Although the detection mechanisms are already known, however, the use of nanoparticles polymer composites based on MPA-Fe_3_O_4_/NCC/CTAB with the use of Ru(bpy)_3_^2+^ as an electrochemical indicator has not yet been reported in any MTB assay. Therefore, the present study can also be considered as improving knowledge in the development of electrochemical DNA sensor. Furthermore, their interaction with the Ruthenium complex was able to differentiate between complementary, non-complementary and mismatch DNA for better sensitivity and selectivity of MTB detection.

## 2. Results

### 2.1. The Field Emission Scanning Electron Microscopy (FESEM) Analysis

FESEM results revealed that the morphology of the Fe_3_O_4_ in Figure 1A shows an agglomerated and a spherical shape. Interestingly, functionalization with MPA has changed its structure to the bundling structure of nanowire compared to uncapped Fe_3_O_4_ (Figure 1B). Subsequently, in Figure 1C, FESEM image shows NCC morphology in the presence of CTAB displayed a nanocrystal aggregated in the form of bundles with a width of approximately 50 nm and a length of 200 nm.

### 2.2. Electrochemical Characterization

To evaluate the interface properties of modified electrode, cyclic voltammetry (CV) was used in an electrochemical system to study acceleration electron transfer between the electrochemical probe and the electrode. In this study, CV experiments were carried out in 5 mM of (Fe(CN)_6_^3^/Fe(CN)_6_^4^) in 0.1 M KCl consist of five different modified electrodes of (a) bare SPCE, (b) SPCE/Fe_3_O_4_, (c) SPCE/MPA-Fe_3_O_4_, (d) SPCE/NCC/CTAB, and, (e) SPCE/MPA-Fe_3_O_4_/NCC/CTAB with scan rate 100 mV/s (Figure 2). The bare electrode of SPCE (curve a) exhibited a well-defined oxidation and reduction peaks in which the semi-reversible curve of Fe(CN)_6_^3^/Fe(CN)_6_^4^ peak was observed. Modification SPCE with Fe_3_O_4_ (curve b) showed an increase in the anodic peak current from 70.68 to 99.60 μA. Interestingly, functionalization of Fe_3_O_4_ with MPA on the SPCE (curve c) showed raises of current, 22.40 μA. On the other hand, after the modification with cationic surfactant of CTAB, as a nonconductive material, NCC displayed increasing peak current (curve d). Subsequently, after deposition of MPA-Fe_3_O_4_/NCC/CTAB on SPCE, a large increase in peak current (177.18 µA) was observed. For comparison, bare SPCE has been studied using the same basic protocol. In this case, the CVs in 5.0 mM [Fe(CN)_6_]^4−/3−^ solution containing 0.1 M KCl at different scan rates were used. The electroactive surface area of SPCE/MPA-Fe_3_O_4_/NCC/CTAB modified electrode was calculated based on the Randles–Sevcik equation ip = 2.69 × 10^5^ n^3^/2AD^1/2^Cv^1/2^, where ip = maximum current (ampere), n = number of electron transfer (in this work n = 1), D = diffusion coefficient (cm^2^/s) of [Fe(CN)_6_]^3−/4−^ solution (7.6 × 10^−6^ cm^2^/s), A = electrode area (cm^2^), C = concentration (mole·cm^−3^) and v = scan rate (mV/s) [29]. As a result, the calculation electroactive surface area of the SPCE/Fe_3_O_4_/NCC/CTAB modified electrode to be 0.167 cm^2^ as compared to the bare SPCE of 0.076 cm^2^. Moreover, the additional information from the plotted graph of anodic and cathodic peak currents against the square root of the scan rate displayed the linearity of the plots (R^2^ = 0.998).

### 2.3. Electrochemical Performance of Fabricating Electrochemical DNA Biosensor

Figure 3 shows current measurements in the presence of Ru(bpy)_3_^2+^, which consist of SPCE/MPA-Fe_3_O_4_/NCC/CTAB/EDC/NHS (curve a), SPCE/MPA-Fe_3_O_4_/NCC/CTAB/EDC/NHS/DNA (curve b), SPCE/MPA-Fe_3_O_4_/NCC/CTAB/EDC/NHS/DNA-DNA (curve c), respectively. In the absence of DNA, Ru(bpy)_3_^2+^ displayed the oxidation signal at 1.1 V and reduction signal at 0.8 V (curve a).

### 2.4. Selectivity and Sensitivity of the Developed Electrochemical DNA Sensor

Subsequently, the selectivity property of developed biosensor (curve a) was further investigated with single stranded DNA (ssDNA) (curve b), mutation DNA (curve c), non-complementary DNA (curve d) and complementary DNA (curve e) using DPV analysis as shown in Figure 4. It is pertinent to note that the peak current produced esulted from the Ru(bpy)_3_^2+^ oxidation peak. As can be seen, the lowest peak current was observed on the DPV signal in the absence of DNA target on the modified electrode (curve a). Furthermore, various concentrations of ssDNA (1.0 × 10^−6^ M to 1.0 × 10^−12^ M) were initially hybridized on the modified electrode. Subsequently, the current changes in DPV were recorded in Ru(bpy)_3_^2+^. Comparison between previous electrochemical studies using various nanomaterials and biomolecules, based on the screen-printed electrode for tuberculosis detection, is summarized in Table 1.

### 2.5. Detection of PCR Product of Mycobacterium Tuberculosis

Several samples of *Mycobacterium tuberculosis* from PCR amplified product was tested as DNA target by using the developed electrochemical biosensor for applicability studies. In Figure 5A, the result from analyzed agarose gel electrophoresis separation showed a fragments band at 157 bp was an indication of positive results related to MTB. Each line from agarose gel represent positive control (D1), negative control (D2), negative MTB sample (D3), MTB samples (D4, D5, D6 and D7), and *Mycobacterium* other than tuberculosis (MOTT), such as *Staphylococcus aureus* (D8), respectively. Subsequently, the same sample was evaluated by the fabricated DNA biosensor for real time setting, and the current produced was compared, as presented in Figure 5B,C. As a result, the DPV response showed an increase in the peak current using a positive control sample (Curve D1) and MTB positive samples (Curve D4, D5, D6, and D6) compared to ssDNA probe (Curve D0).

## 3. Discussion

### 3.1. The Field Emission Scanning Electron Microscopy (FESEM) Analysis

Figure 1A exhibited a result which is similar to the observation reported by previous study [33]. The bundling structure (Figure 1B) suggests the other way around forming Fe_3_O_4_ slender structures like nanowire array without the need for any precipitation and a subsequent microwave hydrothermal process, as reported in the previous study [34]. The agglomeration of the nanocrystals (Figure 1C) might be due to the evaporation of water during the sample preparation [35]. Subsequently, the morphology of the resulting composites of MPA-Fe_3_O_4_/NCC/CTAB showed the bulky nanoparticle agglomerated together (Figure 1D). This could be due to intermolecular forces that mediated the interaction between the molecules of the nanoparticles.

### 3.2. Electrochemical Characterization

The modified SPCE with Fe_3_O_4_ (curve b) was able to facilitate the electron transfer kinetics in order to offer much higher conductivity [36]. Furthermore, the functionalization of Fe_3_O_4_ with MPA (curve c) was able to assist electron acceleration to the electrode surface. On the other hand, the SPCE modification with CTAB and NCC exhibited the synergistic effect between MPA-Fe_3_O_4_ and NCC/CTAB to facilitate the electron transfer kinetics of Fe(CN)_6_^3^/Fe(CN)_6_^4^ (curve e). In addition, a further experiment was performed to calculate the real electroactive area, an important parameter to be considered in the development of electrochemical DNA biosensor. The results from CV demonstrate that CTAB was able to provide conductive properties to NCC in order to accelerate the electron transfer on the electrode surface. As previously reported, the introduction of the CTAB surfactant has increased the dispersion of cellulose nanocrystals (CNCs) and the dispersive effect of nanoparticles increases the conductive probability [27,37]. The results also show that the electroactive surface area of SPCE/MPA-Fe_3_O_4_/NCC/CTAB was increased by approximately 54.5% compared to the bare SPCE. This increment shows the ability of the developed electrochemical for providing a wide area platform [32] for DNA immobilization; thus, it can influence the sensitivity of detection [18,38]. Furthermore, the linearity of the plots has suggested that the electron transfer occurred was due to the diffusion-controlled electron transfer process [27,39].

### 3.3. Electrochemical Performance of Fabricating Electrochemical DNA Biosensor

The use of Ru(bpy)_3_^2+^ as the electrochemical indicator was expected to be able to distinguish between single stranded DNA (ssDNA) and double stranded DNA (dsDNA) to achive a highly sensitive response from the developed biosensor. As previously reported, there are at least three DNA-binding modes with metal complexes available, consisting of electrostatic attraction, intercalation, and groove binding [40,41]. In this study, the interaction between positive charges of Ru(bpy)_3_^2+^ and negative charges of DNA could occur with combinations of all of these modes. Furthermore, the increase in the peak current after the DNA hybridization suggested that Ru(bpy)_3_^2+^ binds to the groove of the DNA duplex in which the interaction occurred between the Ru(bpy)_3_^2+^ and the phosphate backbone, indicating the intercalation of long-range guanine oxidation in DNA duplexes [42].

### 3.4. Selectivity and Sensitivity of the Developed Electrochemical DNA Sensor

The drastically increased DPV peak current was observed after DNA immobilization on the modified electrode indicating successful Ru(bpy)_3_^2+^ intercalation into the DNA (curve b). Furthermore, the highest peak current was shown after hybridization with the complementary target DNA (curve e), indicating that a large amount of Ru(bpy)_3_^2+^ interacted with dsDNA through the binding modes mechanism, as mentioned in Section 3.3. Moreover, it is apparent from this study that the binding between DNA and mutation DNA (curve c) resulting in an insignificant increase in current due to non-specific binding. This result shows the ability of the developed electrochemical biosensor to discern the DPV signal between complementary DNA, mutation DNA, and non-complementary DNA in the presence of Ru(bpy)_3_^2+^. The results also show that the oxidation peak increased linearly with the increased concentration of complementary target DNA correspond with the linear calibration curve obtained in Figure 4B [Y = 0.6936 log (X) + 11.47] where Y is the peak current and log (X) is the logarithm of the target DNA concentration with a correlation coefficient of R^2^ = 0.9896. Meanwhile, the limit of detection (LOD) was calculated based on 3σ/S [18] where σ is the relative standard deviation (RSD) of blank (n = 3) and S represents slope of calibration curve with the LOD obtained was 7.96 × 10^−13^ M. The reproducibility of the fabricated DNA sensor was evaluated by using an eight replicate system with the RSD value calculated to be at 3.87% (n = 8). The results indicate that the method is consistent within the error limit and meaningful for the designed biosensor. Furthermore, the stability study of the fabricated DNA sensor displayed a good result after it was stored at 4 °C for 7 days with minimum loss of activity. The comparison results from Table 1 clearly indicate that the developed DNA biosensor displayed a good analytical performance if compared to the previous reported electrochemical biosensor in terms of the limit of detection.

### 3.5. Detection of PCR Product of Mycobacterium Tuberculosis

The developed electrochemical biosensor was exposed to PCR product from real sample MTB. Compared to traditional PCR, the developed electrochemical biosensor does not require a step of visualization of amplified products by gel electrophoresis, which may be time-consuming and costly. Following the preparation step for electrode hybridization, as mentioned in Section 4.9, it can be seen that the higher peak current produced resulted from complementary binding between the ssDNA probe and MTB-positive sample on the modified electrode. Meanwhile, the DPV response for D2, D3, and D8 samples resulted in a decreased current response compared to D1, D4, D5, D6 and D7 sample indicated to the lack of hybridization occurred between DNA probe and PCR product. In the absence of the hybridization case, this is possibly due to the non-specific interaction, which is in agreement with the DPV analysis using synthetic DNA (Figure 4A). The results from this study show that a significant DPV response was obtained to distinguish a signal between positive and negative MTB and, therefore, the fabricated electrochemical DNA sensor demonstrated good potential for selectivity and effectively detection toward the MTB from the PCR product.

## 4. Materials and Methods

### 4.1. Reagents

Iron oxide (Fe_3_O_4_) was obtained from Materials Synthesis and Characterization Laboratory, Institute of Advanced Technology, Universiti Putra Malaysia (ITMA). Tris (2,2′-bypridyl) dichlororuthenium(II) hexahydrate (Ru(bpy)^2+^), cetyltrimethylammonium bromide (CTAB), ethylenediaminetetraacetic acid (EDTA), Tris HCI NH_2_C(CH_2_OH)_3_ ⋯ HCl, potassium ferricyanide (K_3_[Fe(CN)_6_]), and N-hydroxysuccinimide (NHS) were obtained from Sigma-Aldrich (Steinheim, Germany). Cellulose nanocrystalline (NCC) was purchased from the Process Development Centre, University of Maine. N-ethyl-N-(3-(dimethylaminopropyl) carbodiimide (EDC) was bought from Fluka (Switzerland). 3-mercaptopropanoic acid (MPA) was purchased from R & M chemicals. All other chemicals were of analytical reagent grade. Meanwhile, synthesized oligomers were purchased from 1st Based Laboratory Sdn. Bhd., Malaysia and based on sequences listed in Table 2 below:

### 4.2. Real Sample Application

The genomic DNA of the MTB was extracted from the sputums of patients who were diagnosed with TB (positive), which were provided by the Institute for Medical Research (IMR), Malaysia. In order to amplify the obtained products, polymerase chain reaction (PCR) was carried out using the mixture of primers, as stated in Table 1. After 30 cycles of PCR, the products obtained were identified by running 5 µL of a PCR mixture in 1.0% agarose gel for 45 min and were examined under ultraviolet light. The agarose gel electrophoresis for the positive TB sample revealed products at fragment sizes of 157 bp, while no band appears to indicate a negative result. Meanwhile, denatured single-stranded DNA (ssDNA) was obtained by heating dsDNA in a water bath between 50 and 60 °C for 10 min, followed by rapid cooling in an ice bath.

### 4.3. Preparations of Stock Solutions

DNA oligomers (100 µM) were prepared in Tris EDTA (TE) buffer solution containing 10 mM Tris-HCI and 1 mM EDTA (pH 8.0), labeled as “stock solution” and kept frozen at −20 °C. When required, probe DNA, target DNA, non-complimentary DNA and mutation DNA were defrosted, and diluted concentrations were prepared using TE buffer containing 20 mM NaCI.

### 4.4. Instrumentation

Screen printed carbon electrodes (SPCEs) were purchased from Dropsens (Asturias, Spain), and consisted of the working electrode (carbon), reference electrode (Ag), and auxiliary electrode (carbon). Voltammetric measurements were obtained using the Autolab (Ecochemie, Utrecht, The Netherlands) potentiostat incorporated with a general-purpose electrochemical system (NOVA 1.11).

### 4.5. Preparation of MPA-Fe_3_O_4_

The capping process was based on previous work with slight modification [28]. Firstly, 1 wt% of Fe_3_O_4_ was mixed with MPA using a mole ratio of 1:30 in deionized water and shaken for overnight. The particles were washed and dried at room temperature. MPA-Fe_3_O_4_ (5 mg) was dissolved with 1 mL deionized water and mixed.

### 4.6. Preparation of NCC/CTAB

Modification of NCC with CTAB was prepared as previously described with slight modifications [43]. Firstly, 1 g NCC (1 wt% suspension at pH 10 adjusted using 1 M NaOH) along with 1 wt% aqueous CTAB solution with 2:1 ratio between CTAB to sulfur was prepared. The suspension of NCC was added slowly to the CTAB solution, and the foamy mixture was maintained at 60 °C for 3 h and left stirring overnight at room temperature. Afterwards, the samples were washed to remove the unbound CTAB by centrifugation at 10,000 rpm for 10 min.

### 4.7. Preparation of MPA-Fe_3_O_4_/Ncc/Ctab

The solution of MPA-Fe_3_O_4_ and NCC/CTAB was mixed well-using volume ratio of 1:1. Prior to the SPCE modification, 7 µL MPA-Fe_3_O_4_/NCC/CTAB was prepared using drop casting method on the SPCE and dried at room temperature overnight.

### 4.8. Immobilization of DNA Probe onto Modified SPCE/MPA-Fe_3_O_4_/Ncc/Ctab

The modified SPCE was immersed in 5 mM EDC and 2 mM NHS containing 50 mM PBS (pH 5.2) for 10 min at room temperature. Subsequently, 2 µL of probe DNA (10 µM) was dropped onto the surface of modified SPCE and incubated at room temperature for 1 h. The electrode was then washed with TE buffer for 30 s. The washed electrode was denoted as SPCE/MPA-Fe_3_O_4_/NCC/CTAB/EDC/NHS/DNA throughout.

### 4.9. Hybridization of DNA

Subsequent process was performed by hybridized 2 μL of complementary ssDNA on SPCE/MPA-Fe_3_O_4_/NCC/CTAB/EDC/NHS/DNA probe as above with desired concentration and incubated for 25 min at room temperature. The successful hybridize complimentary DNA on the modified SPCE was washed with TE buffer (pH 7) and purged with nitrogen gas (N_2_). Subsequently, electrochemical signals of ruthenium bypridyl (Ru(bpy)_3_^2+^) bound to signal DNA were measured by differential pulse voltammetry (DPV). Hybridization was monitored using different target DNA concentrations ranging from 1.0 × 10^−7^ to 1.0 × 10^−13^ M, followed by non-complimentary DNA, mutation DNA, and DNA from clinical samples (from IMR). The fabrication procedure of the DNA electrochemical biosensor can be viewed as illustrated in Scheme 1.

## 5. Conclusions

The present study demonstrated a sensitive electrochemical DNA biosensor platform based on a new composite nanomaterial. The combination of MPA-Fe_3_O_4_ and NCC/CTAB was successfully prepared for the immobilization of the capture probe to be hybridized with the complementary target DNA of MTB. Detection of complementary target DNA ranging from 1.0 × 10^−12^ to 1.0 × 10^−6^ M was obtained with a limit of detection of 7.96 × 10^−13^ M. The DPV analysis showed that the fabricated DNA electrochemical biosensor exhibited high sensitivity and specificity with the synthetic DNA and also the clinical sample from the PCR product (MTB). The developed biosensor also demonstrated a differentiation to the interferences such as negative MTB, *Mycobacterium* other than tuberculosis, and the respiratory-related bacterial sample such as *Staphylococcus aureus.* These characteristics make the fabricated biosensor an attractive alternative for the in site application and can provide a promising platform for rapid diagnosis of tuberculosis.

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
