# Peer review of "DNA Electrochemical Biosensor Based on Iron Oxide/Nanocellulose Crystalline Composite Modified Screen-Printed Carbon Electrode for Detection of Mycobacterium tuberculosis"

_molecules, 2020, doi:10.3390/molecules25153373_

Round 1

Reviewer 1 Report

This manuscript describes a DNA electrochemical biosensor for the detection of mycobacterium tuberculosis (MTB), an infectious bacterium responsible of tuberculosis (TB) pulmonary diseases. Iron oxide/nanocellulose crystalline composite is used to modify a thick-film screen printed carbon electrode for the detection of MTB. Super-paramagnetic iron oxide (SPIO) nano-particles have been used by other researchers in tuberculosis diagnosis. 3-mercaptopropinic acid (3-MPA) is used to functionalize Fe3O4, and then composited with nanocellulose crystalline, Cetyltrimethylammonium Bromide, and then onto the thick-film printed carbon electrode. These preparation steps are logical. A minor clarification may be need by the authors. In the Materials section, Hexadecylltrimethylammonium Bromide, is listed which differs from Cetyltrimethylammonium Bromide. A clarification will be helpful in the chemical actually used. Ru(bpy)3+ is used as the electrochemical indicator, and differential pulse voltammetry (DPV) is the transduction mechanism of this biosensor.

TB is caused by a bacterium, mycobacterium tuberculosis (MTB). MTB also attacks other organs in the bogy, including lung, kidney and others. Thus, the detection of MTB is clinically meaningful. Most of the MTB diagnostic methods involve skin test, acid-fast and fluorescent stains as well as polymerase chain reaction (PCR) which are time-consuming and require skillful operator. Therefore, a simple biosensor for the detection as described by this manuscript is practical aiming toward to minimize the complication of the current MTB diagnostic techniques.   The description in preparing this DNA electrochemical based biosensor in this manuscript is reasonably sound and doable. This is the positive aspect of this manuscript. Nevertheless, the preparation steps of this biosensor is relatively elaborated. It is suggested that the authors should remind the readers about the elaboration steps involved.

Figure 1 of the manuscript shows the FESEM image of the morphology of Fe3O4 in different steps of the biosensor preparation. The authors may want to show the image of the electrode after the final binding step with Ru(bpy)3+, even though this linkage may not change the morphology of the electrode. The author may also just make a statement indicating the final binding of the electrochemical indicator, Ru(bpy )3+ does not change the electrode surface structure.

Line 307 of p.10 of this manuscript, the word “electrode” does not needed to be capitalized.

Reviewer 2 Report

This work on the detection of TB by an electrochemical method based on screen-printed electrodes and MPA-Fe3O4/NCC/CTAB is interesting and all the needed analytical parameters have been well analysed.

Please, find here below some minor comments:

-in the Introduction, the authors claim “Furthermore, the implementation of nucleic acid amplification assays such as PCR is limited by high cost and poor performance under extreme field conditions [9]. Therefore, there is a need to develop another fast screening method for in situ and real-time analysis.”. However, to validate their method with real samples they use PCR amplified DNA, so they still rely on the amplification by PCR. Please, add a discussion on this point;

-when making hybridization among the immobilized probe and the PCR sample, usually the PCR sample is treated in order to produce a single stranded DNA filament since the product of the PCR amplification is a double stranded DNA fragment. Please, specify how the PCR sample was treated before hybridsation with the probe.
